# Effects of Corner Set−Backs on Wind Loads and Wind Induced Responses of Rectangular Tall Buildings

**Yi Li** [1,2]**, Jieting Yin** [1,2,*] **and Yan Zhang** [1,2]

1   Hunan Provincial Key Laboratory of Structures for Wind Resistance and Vibration Control,
    Hunan University of Science and Technology, Xiangtan 411201, China
2   School of Civil Engineering, Hunan University of Science and Technology, Xiangtan 411201, China
*   Correspondence: 20010201031@mail.hnust.edu.cn

**Abstract:** In order to investigate the effects of corner set-backs on wind loads and wind-induced responses of rectangular high-rise buildings, pressure measurements were carried out on a benchmark model (CARRC) and four models with different rates (5%, 10%, 15%, 20%) of corner set-backs in a boundary layer wind tunnel. The test results show that the corner set-backs contribute to reducing along-wind loads of the rectangular high-rise building models, and the maximum reduction happens at 10% corner set-back. The across-wind loads decrease as the rate of corner set-back is increasing and the maximum reduction emerges at 20% corner set-back. The RMS accelerations at the top of models also decrease with the increasing of rate of corner set-back in along-wind and across-wind. Through the fitting of test results, empirical formulas for the correlation factors of base moment coefficients of rectangular high-rise buildings with different rates of corner set-back are put forward. Moreover, the correlation factors for the power spectrum densities of base moments are listed at typical frequencies corresponding to the practical tall buildings. The outputs of this paper aim to serve as references for wind-resistant design of similar buildings in strong wind region.

**Keywords:** rectangular high-rise building; corner set-back; wind tunnel test; wind loads; wind-induced responses; correlation factor



## 1. Introduction

As the high-strength materials and innovative construction techniques are applied, high-rise building structures tend to be higher and more flexible. These high-rise buildings designed in strong wind regions will be sensitive to wind excitations. It has always been a problem in wind engineering to control or minimize the wind effect on high-rise buildings under strong winds [1–3]. There are three main approaches to reduce the wind load of high-rise buildings: auxiliary damping devices, structural measures and aerodynamic shape modifications [4]. Aerodynamic shape modifications refer to the adjustment of building geometry at the early stage of design to achieve the purpose of reducing wind effects from the sources [5,6].

Corner set-back treatment is a popular corner modification for aerodynamic performance improvement during wind resistant design for high-rise buildings. Kawai [7] studied the aerodynamic instability of high-rise buildings with corner modifications through aeroelastic model tests, and indicated that a model with corner set-back rate of 5% is most effective in preventing the aeroelastic instability of high-rise buildings. Gu and Quan [8] examined the influence of corner modifications on square high-rise buildings through aeroelastic model tests and high-frequency equilibrium tests, and it was demonstrated that corner set-backs can result in a reduction in the peak of power spectral density in across-wind direction. The impact of different types of aerodynamic modifications on the wind effect of high-rise buildings was summarized by Irwin [9], who pointed out that the corner set-back modification can significantly reduce the wind-induced response along

and across wind directions. Tse et al. [10] proved that the corner set-back modification could reduce base moment more effectively along and across wind directions. In this case, 14 square high-rise building models with corner set-backs were tested in two different simulated wind fields by Zhang et al. [11]. The test results show a significant effect of two types of corner set-backs on the aerodynamic coefficients of base moments and torques, and the best option for reducing the aerodynamic coefficients is a model having a corner set-back rate of 7.5%. Pressure tests have been conducted by Yan et al. [12] in order to analyze the wind loads of corner set-back modifications on the high-rise buildings, the test results shown that a 10% corner set-back rate could best reduce the mean and RMS force in an along-wind direction. Alminhana et al. [13] argued that the model with corner set-back generally has a better performance comparing the chamfered corners model in terms of lift force coefficient. Gaur and Raj [14] revealed that building with corner set-back showed better performance to reduce drag force by numerical simulation. Although several research works have been conducted on corner set-backs on the reduction of wind loads and wind induced responses in tall buildings, further studies are still needed on this topic. On the one hand, the studies in the past mainly concentrated on the wind load characteristics of square high-rise buildings with corner set-back, and the associated findings are likely to be inappropriate for rectangular high-rise buildings duo to the difference in aspect ratios. On the other hand, the existing conclusions are lack of empirical formulas considering the effects of different corner set-backs, which can be applied to the assessment of wind effects reductions of high-rise buildings directly for engineering applications. Therefore, there is a need to systematically study the effects of different corner set-backs on the wind loads and wind-induced responses of rectangular tall buildings.

In this paper, the benchmark model is the CAARC standard model, four other similar models with different corner set-back rates are designed accordingly. Wind loads and wind induced responses of rectangular high-rise buildings with various corner set-backs studied by conducting a number of tests. Distributions of wind pressure coefficients, local wind force coefficients, base moment coefficients and the corresponding power spectrum originating from wind tunnel test results are analyzed. Furthermore, correlation factors for wind loads reduction considering the effects of different corner set-back rates are discussed and the corresponding empirical formulas are proposed. Finally, an example is adopted to quantificationally evaluate the corner set-backs on wind-induced accelerations of rectangular high-rise buildings. The results of the paper aim to provide a valuable reference for wind resistant optimization of similar high-rise buildings.

## 2. Experiments

The pressure measurement test was carried out in a wind tunnel [15]. Simultaneous acquisition of wind pressure data from each pressure tap on the model was performed using an electronic pressure scanning module, and the sampling frequency was 333 Hz. The sampling time of each collection is set as 60 s, indicating that the sampling length is 20,000 at each pressure tap. According to the Chinese design code [16], spires and roughness elements were combined to simulate suburban terrain. The test was conducted at a geometric scale of 1/300. The mean wind speed profile along height is shown in Figure 1a, and the turbulence intensity profile along height in Figure 1b. A good agreement between the simulated profiles and the Chinese design code is observed. The mean wind speed is approximately 9.5 m/s at reference height of 0.609 m (reference height is the top of the models). The Reynolds number Re calculated in terms of $U_H$ and the width of the models is $9.65 \times 10^4$. In Figure 2, the longitudinal wind speed spectrum at reference height is well in accordance with the von Karman spectrum.

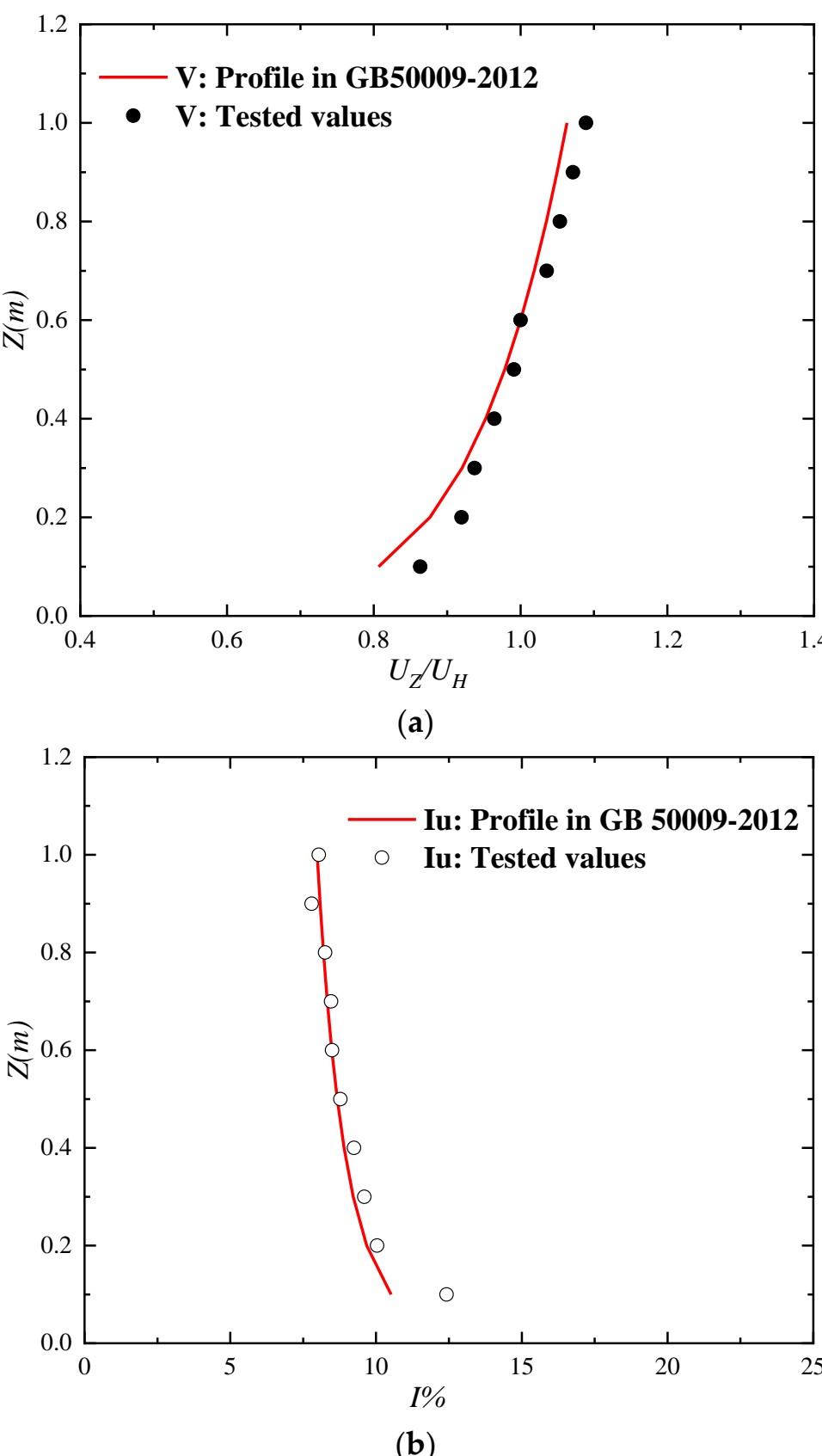

**Figure 1.** The distribution of mean wind speed and turbulence intensity along height. (**a**) The distribution of mean wind speed along height. (**b**) The distribution of turbulence intensity along height.

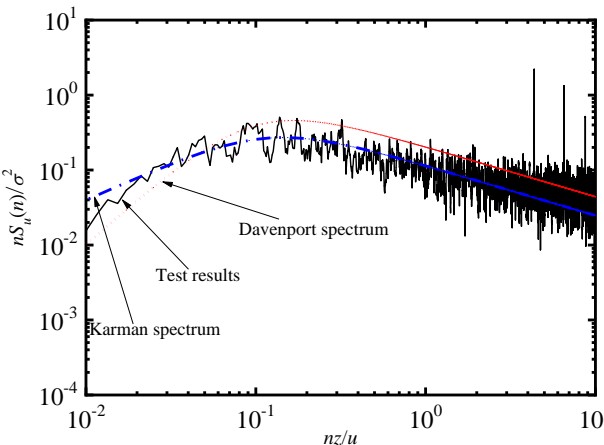

**Figure 2.** The longitudinal wind speed spectrum at the top of the models.

The dimension of benchmark model after scaling is 609.6 mm × 152.4 mm × 101.6 mm (Height × Breadth × Depth), and other models are the benchmark model with different corner set-backs (5%, 10%, 15%, 20%). The rigid models are all made by 5 mm thick ABS (Acrylonitrile Butadiene Styrene) plate, and Figure 3 shows the test model. There are seven measurement layers in each rigid model, and the relative height of each measurement layer is 0.17, 0.33, 0.5, 0.67, 0.80, 0.90 and 0.98, respectively. There are 20 pressure taps in each measurement layer, and a total of 140 pressure taps in a model. Figures 4 and 5 show in detail the arrangement of the pressure taps. The 700 mm length PVC (Poly Vinyl Chloride) tube is used to connect each pressure tap and the scanning module to record the time history of wind pressure. Prior to data processing, the effect of tube is rectified by numerical compensation [17]. The sampling frequency and record length were 333 HZ and 60 s, respectively.

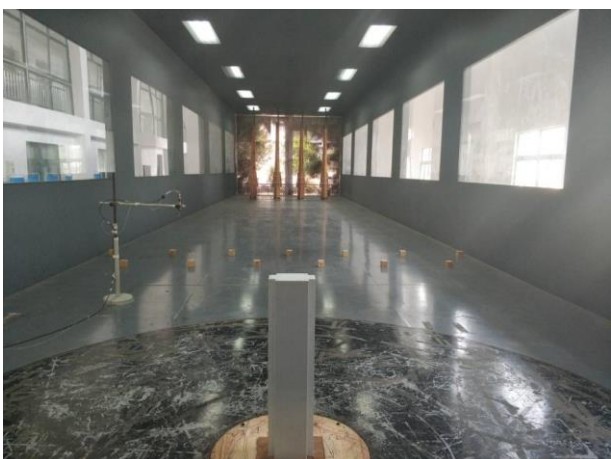

**Figure 3.** The test model.

Assessing the effect of different corner set-backs of rectangular high-rise buildings on wind loads and wind induced response, rate of corner set-back $\gamma$ is defined by the following equation:

$$\gamma = 2b/B \tag{1}$$

In which, $b$ is the corner set-back length and $B$ is the breadth of the benchmark model on windward. The detailed definition of the corner set-back rate is show in Figure 6.

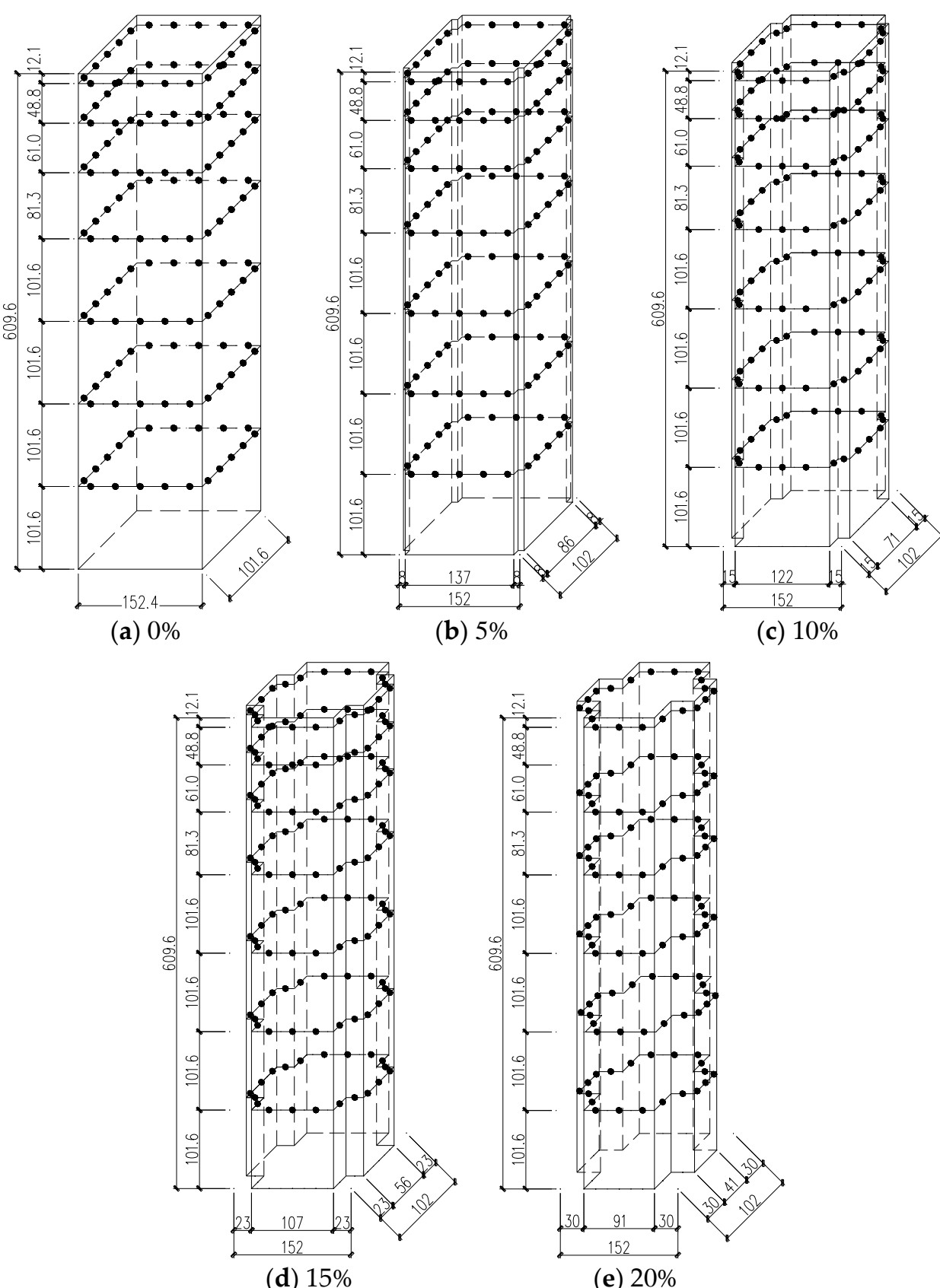

**Figure 4.** The models and pressure tap distributions (Unit: mm).

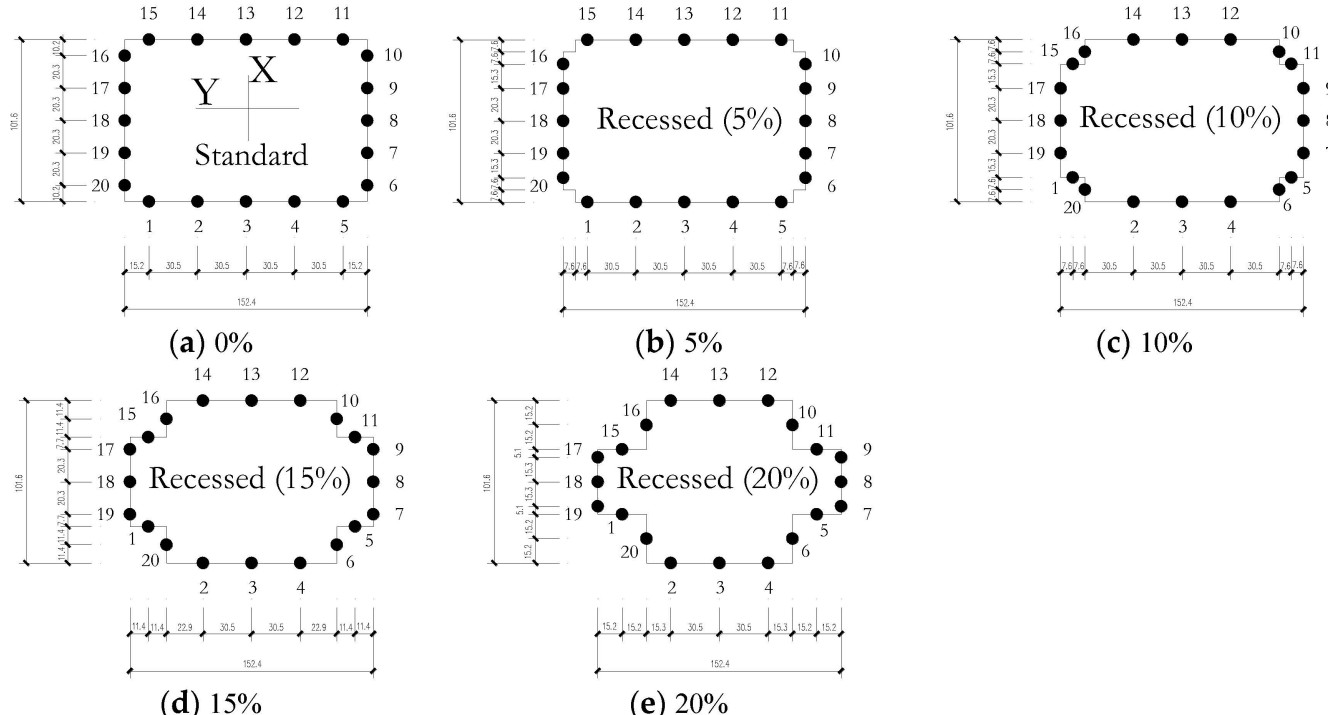

**Figure 5.** The corner set-back modifications and arrangement of measurement taps (Unit: mm).

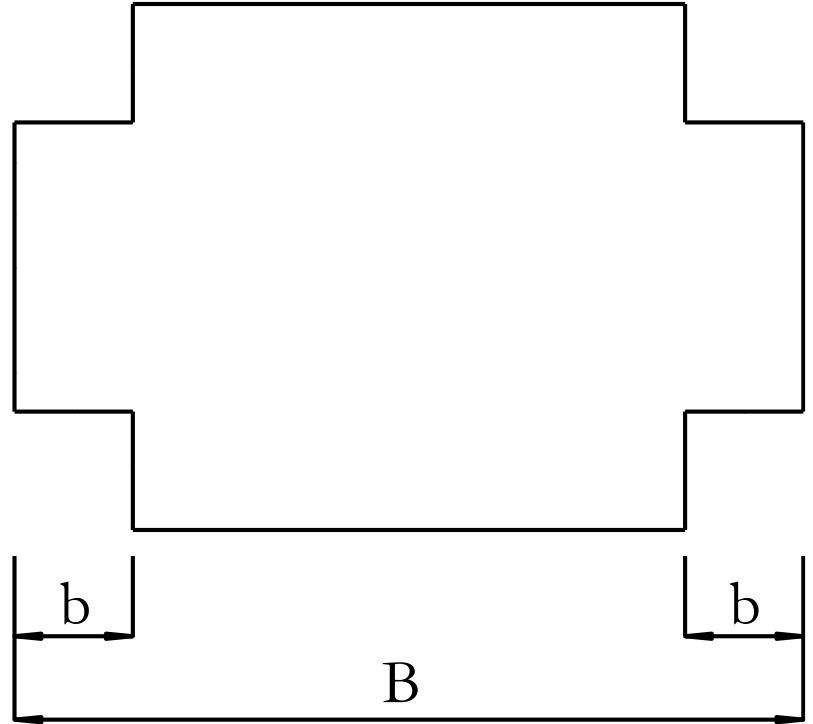

**Figure 6.** The definition of corner set-back (top view).

## 3. Analysis of Test Results

### 3.1. Wind Pressure Coefficients at the Height of 2/3H for Models

The measured pressure is normalized to calculate the wind pressure coefficient at the pressure tap *i* as follows:

$$C_{pi}(t) = \frac{P_i(t) - P_0}{0.5\rho U_H^2} \qquad (2)$$

where $C_{pi}(t)$ is the time history of wind pressure coefficient at the pressure tap $i$, $P_i(t)$ is the time history of wind pressure at the pressure tap $i$, $P_0$ is the static pressure at the reference height of 0.609 m, $\rho$ is the air density and $U_H$ is the mean wind speed at 0.609 m.

The mean wind pressure coefficient $C_{pi,mean}$ is determined with the formula below:

$$C_{pi,mean} = \frac{1}{N}\sum_{t=1}^{N} C_{pi}(t) \tag{3}$$

in which, $N$ is the number of sampling length.

The RMS (root-mean-square) wind pressure coefficient $C_{pi,rms}$ could be calculated by:

$$C_{pi,rms} = \frac{1}{N-1}\sqrt{\sum_{t=1}^{N}\left(C_{pi}(t) - C_{pi,mean}\right)^2} \tag{4}$$

Due to space, Figure 7 only displays the mean and the RMS wind pressure coefficients at the height of 2/3H for all models. It is illustrated in Figure 7a that the mean wind pressure coefficients of tap No. 1 and tap No. 5 on the benchmark model and 5% corner set-back model are positive and symmetric with each other. As the corner set-back rate increases from 10% to 20%, the positions of tap No. 1 and tap No. 5 gradually moves to the corner set-backs, and the mean wind pressure coefficients of these two taps become negative. This is probably due to the influence of the corner set-backs, which reduces the width of the windward side, and leading to an accelerated flow separation at the windward corner. The maximum absolute value of mean negative pressure occurs at 10% corner set-back model. Taps No. 2, No. 3 and No. 4 are located at the center of the windward for all the corner set-back models, so their mean wind pressure coefficients are always positive and change slightly with the increasing of the corner set-back rate. For the leeward, the absolute values of the negative wind pressure coefficients decrease by increasing the rate of corner set-back. This is because the wake width on the leeward narrows due to the increase in corner set-back rate. The mean wind pressure coefficients vary more significantly on the side walls than with the windward and leeward, especially for tap No. 6. The absolute value of the mean wind pressure coefficients at the tap No. 6 and tap No. 20 of the models with the corner set-back rates of 5% and 10% are greater than the benchmark model.

In Figure 7b, the RMS wind pressure coefficients of taps No. 1 and No. 5 of the 10%, 15% and 20% corner set-back models increase sharply. This may arise from the hit of shedding vortex on the leading edge of the windward. The corner set-backs do not affect the RMS wind pressure coefficients on the windward (taps No. 2, No. 3 and No. 4) much. The RMS wind pressure coefficients on leeward gradually decreases as the rate of corner set-back increases. For the side wall, the maximum RMS wind pressure coefficient of the benchmark model happens at tap No. 10, which results from the reattachment of flow. Similar phenomenon can be also observed at tap No. 10 of the 5% corner set-back model. However, with the increasing of the corner set-back rate from 0% to 20%, the RMS wind pressure coefficients of tap No. 10 decrease rapidly. The RMS wind pressure coefficients of taps No.6 and No.20 of the 10%, 15% and 20% corner set-back models are greater than those of the benchmark model. Notably, the 5% corner set-back model has greater RMS wind pressure coefficients on the side wall than other models.

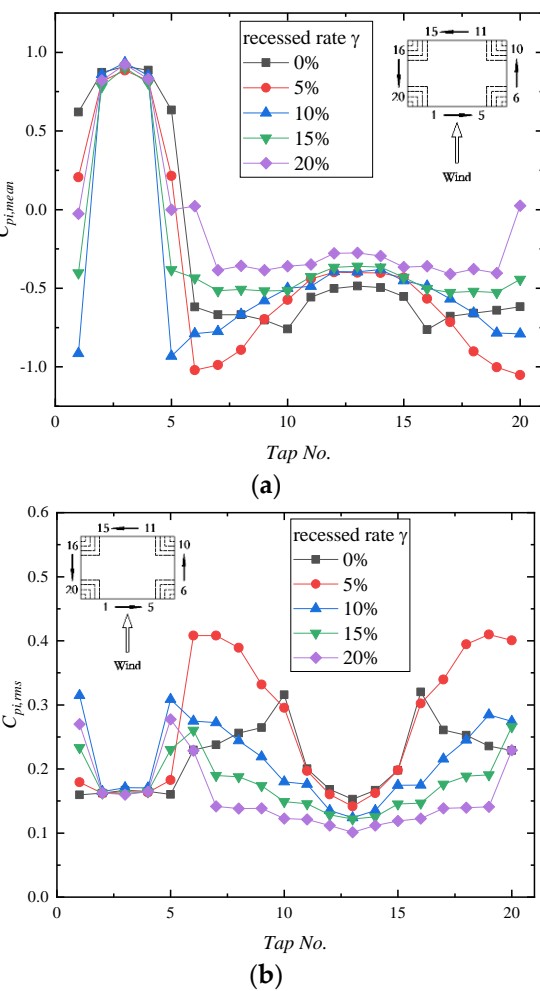

**Figure 7.** The comparisons of wind pressure coefficients at the height of 2/3H for corner set-back models. (**a**) Mean wind pressure coefficients at the height of 2/3H for models. (**b**) RMS wind pressure coefficients at the height of 2/3H for models.

### 3.2. Local Wind Force

Wind force is composed by drag force, lift force and torque. For square or rectangular tall buildings, wind-induced torque is usually very small. Therefore, only the drag force and lift force are discussed in this paper. The mean and RMS drag force coefficients are obtained with Equation (5) and the RMS lift force coefficients are determined by Equation (6):

$$C_D(z_i) = \frac{\overline{F_D(z_i)}}{A(z_i)q_H}, \ C'_D(z_i) = \frac{\sigma_D(z_i)}{A(z_i)q_H} \tag{5}$$

$$C'_L(z_i) = \frac{\sigma_L(z_i)}{A(z_i)q_H} \tag{6}$$

where, $C_D(z_i)$ is the mean drag coefficient of the $i$ th measurement layer, $C\prime_D(z_i)$ and $C\prime_L(z_i)$ are the RMS drag coefficient and RMS lift coefficient of the $i$ th measurement layer, respectively, $F_D(z_i)$ is the mean drag force integrated by the simulated measured wind pressures from pressure taps and the associated area on the $i$ th measurement layer, $\sigma_D(z_i)$ and $\sigma_L(z_i)$ are the corresponding RMS drag force and RMS lift force, $A(z_i)$ is building frontal area of the $i$ th measurement layer, and $q_H$ is the incoming wind pressure at the top of the model.

Figure 8 illustrates the definition of wind force and wind attack angle. Since the unfavorable wind direction for wind resistant design of an isolated rectangular tall building

is 0° [18], and the mean lift force and the mean torsional component under this attack angle are close to zero, hereby only the mean drag force, RMS drag force and RMS lift force of the five models are discussed in this study.

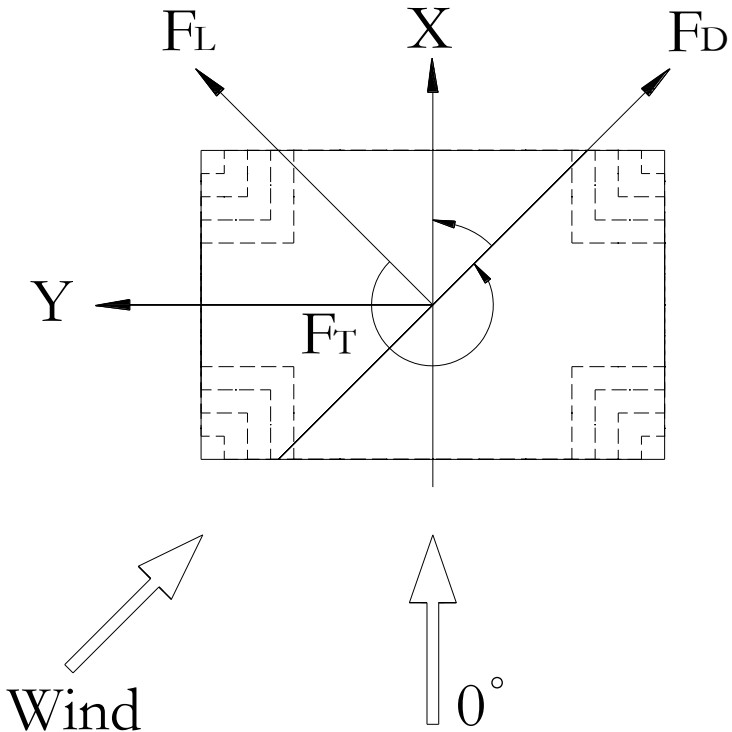

**Figure 8.** The definition of wind forces and wind attack angle.

3.2.1. Local Wind Force Coefficients

Figure 9 illustrates the variations of the local wind coefficients for five corner set-back models along the height. In Figure 9a, the mean drag coefficients gradually increase along the model height and decrease sharply at the top of model, due to the effects of the three-dimensional flow [19]. The corner set-back modification can reduce mean drag coefficients of rectangular tall building model at different extent. The maximum reduction of the mean wind force is found when the rate of corner set-back is 10%. In Figure 9b, the RMS drag force coefficients are also reduced by the corner set-back modifications, which is because the RMS wind pressure coefficients on the leeward are reduced. Once the rate of corner set-back exceeds 5%, the reductions caused by corner set-backs are almost the same for different rate of corner set-back including 10%, 15%, and 20%. It is clear in Figure 9c that the RMS lift coefficient increases and then decreases along the height of the model, and the maximum value usually occurs around 0.7 H. Compared with the benchmark model, the RMS lift coefficients of other models are basically smaller than those of the benchmark model, except for the 5% corner set-back model, which is because the lateral RMS wind pressure coefficients are reduced. The overall RMS life coefficient decreases as the rate of corner set-back increases. This probably relates to the reduction of the separated shear layer thickness on the side walls caused by the corner set-back modifications. The RMS lift coefficients for the 5% corner set-back model are greater than those of the benchmark model at the middle height. This may be due to the formation of vortices on the corner set-back of the side walls.

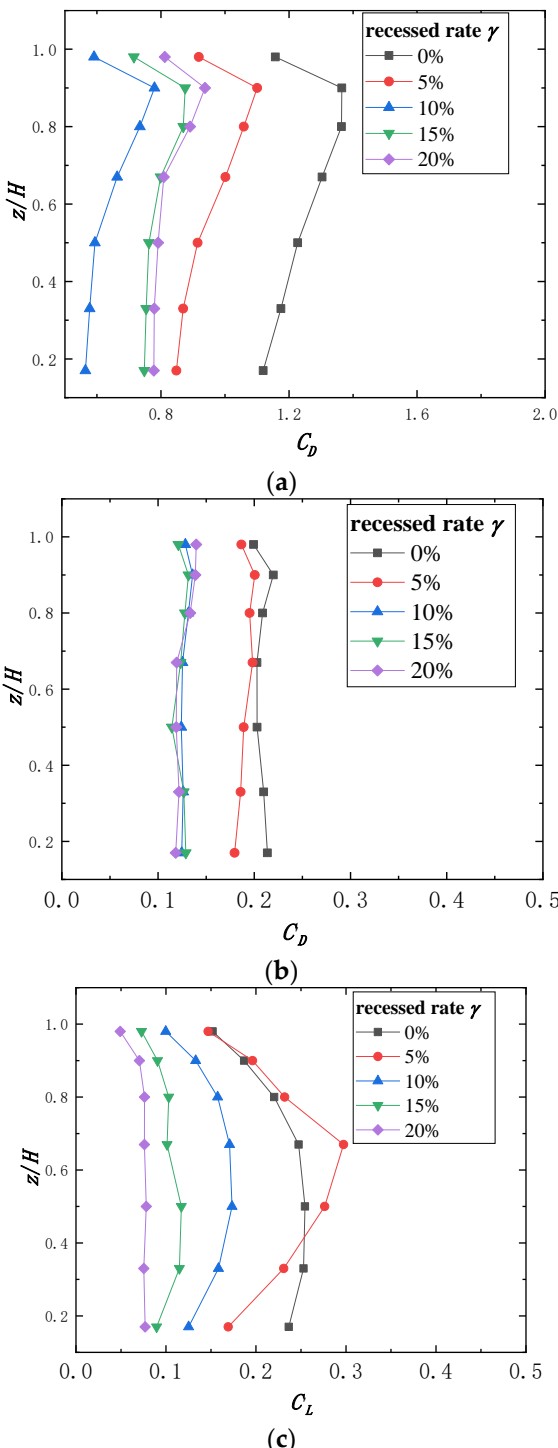

**Figure 9.** The local wind force coefficients. (**a**) Mean drag force coefficients along height. (**b**) RMS drag force coefficients along height. (**c**) RMS lift force coefficients along height.

### 3.2.2. Power Spectral Densities of Local Wind Force Coefficients

Figure 10 presents the power spectra densities of local wind force coefficients at 2/3 H of the five models. As shown in Figure 10a, the power spectra densities of drag force coefficients vary gently with the reduced frequency. Corner set-backs have little effects on the power spectra densities of drag force. From Figure 10b, it can be seen that as the reduced frequency is approaching the Strouhal number of 0.10, the spectrum of the benchmark model has a spike. Although narrow-band characteristics are still presented in the power spectra of lift forces of 5%, 10%, 15% and 20% corner set-back models, their peak values are

lower than that of the benchmark model at different extent, which indicates that the corner set-backs can weaken the energy of the vortex shedding. The power spectra of 20% recessed model have the lowest peak and the widest frequency band. It is also found that the peak values for the power spectra of 10% and 15% corner set-back models are similar to that of the benchmark value, even larger than that of the 5% and 20% corner set-back models.

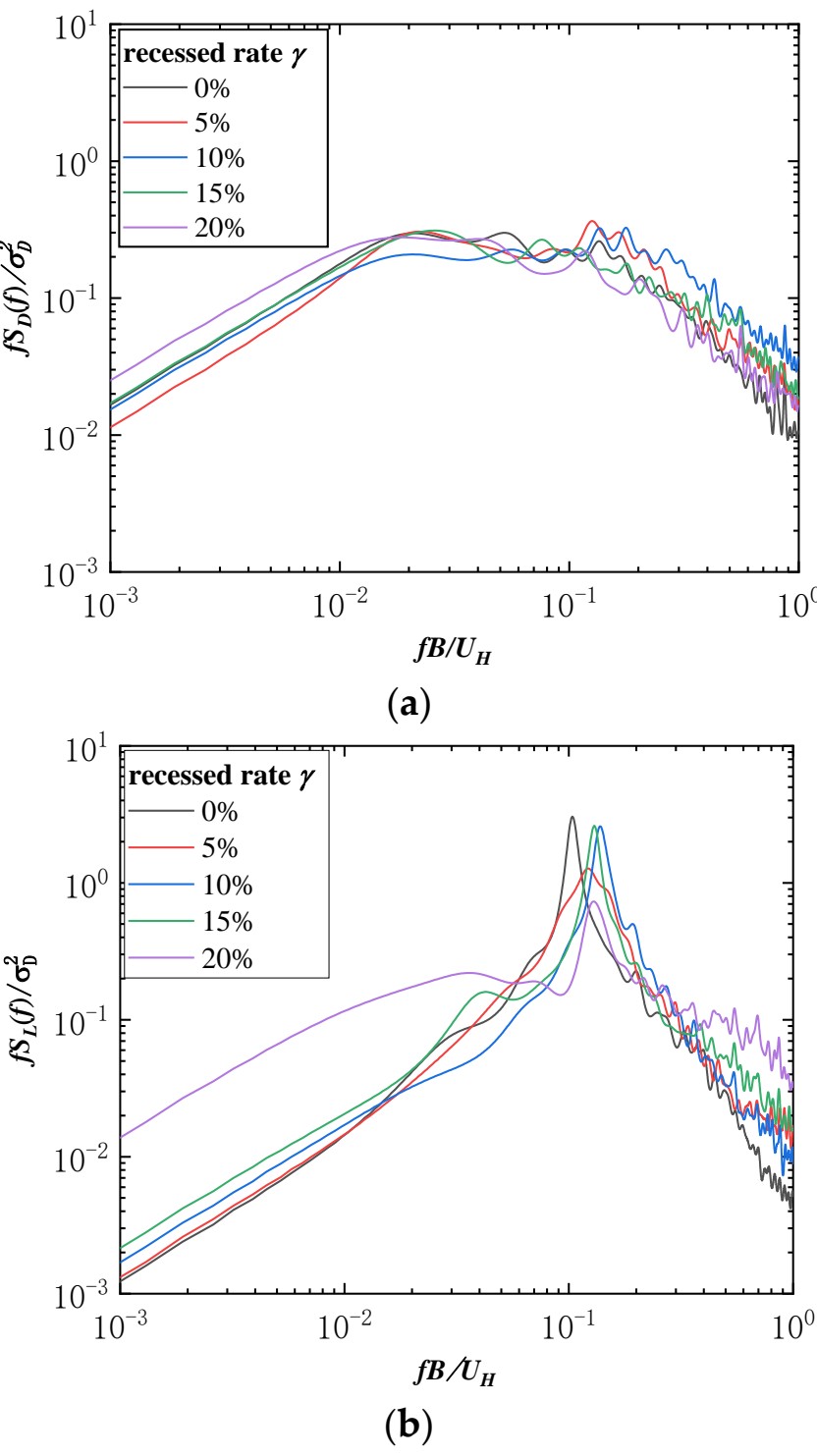

**Figure 10.** The power spectra of local wind force coefficients at 2/3 H of the five models. (**a**) The power spectra of drag force coefficients. (**b**) The power spectra of lift force coefficients.

### 3.3. Base Moment

3.3.1. Base Moment Coefficients

Similarly, only the mean and RMS base moment coefficients at along-wind direction, the RMS across-wind base moment coefficients are analyzed in this paper. The corresponding base moment coefficients are calculated with following equations:

$$C_{MD} = \frac{\overline{F_{MD}}}{BH^2 q_H}, \ C'_{MD} = \frac{\sigma_{MD}}{BH^2 q_H} \tag{7}$$

$$C'_{ML} = \frac{\sigma_{ML}}{BH^2 q_H} \tag{8}$$

in which, $C_{MD}$ is mean base moment coefficient in along-wind, $C'_{MD}$ and $C'_{ML}$ are the RMS along-wind and across-wind base moment coefficients, respectively. $\overline{F_{MD}}$, $\sigma_{MD}$ and $\sigma_{ML}$ are the mean base moment in along-wind, RMS base moment in along-wind and across-wind, respectively. $B$ is the breadth of the windward of the model, $D$ is the depth of the side ward of the model, and $H$ is the height of the model.

Figure 11 presents the variations of base moment coefficients with the corner set-back rate. Since the aspect ratio (Depth to Breadth) of the benchmark building model is 2/3, several outcomes from previous studies [15,20,21] and design codes [16,22] on the base moment coefficients of this typical tall building have also been listed for comparison. From Figure 11a, it can be found that as the rate of corner set-back increases, the mean along-wind base moment coefficient first decreases and then increases. The minimum value 0.338 was observed when the rate of corner set-back is 10%. This variation trend is in line with the trend of the mean drag force. The RMS base moment coefficients of along-wind direction decrease as corner set-back rate increases, although there is a tiny increase when the rate of corner set-back is 20%. When the rate of corner set-back exceeds 5%, the RMS along-wind base moment coefficients nearly reach stable state and keep at about 0.051. With regard to the RMS across-wind base moment coefficients, it is found from Figure 11c that the stipulated values from the Chinese design code [16] and suggested values from the previous studies [20,21] are conservative. The value provides by the Japanese design code is more reasonable to describe the across-wind loads. The RMS base moment coefficient in across-wind decreases from 0.102 to 0.031 when the rate of corner set-back increases from 0% to 20%, indicating that larger rate of corner set-back could be effective in reducing the loads of rectangular high-rise buildings in across-wind.

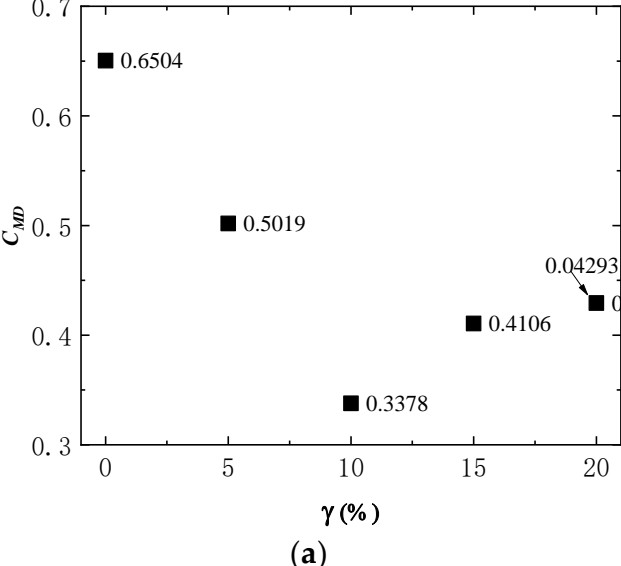

**(a)**

**Figure 11.** *Cont.*

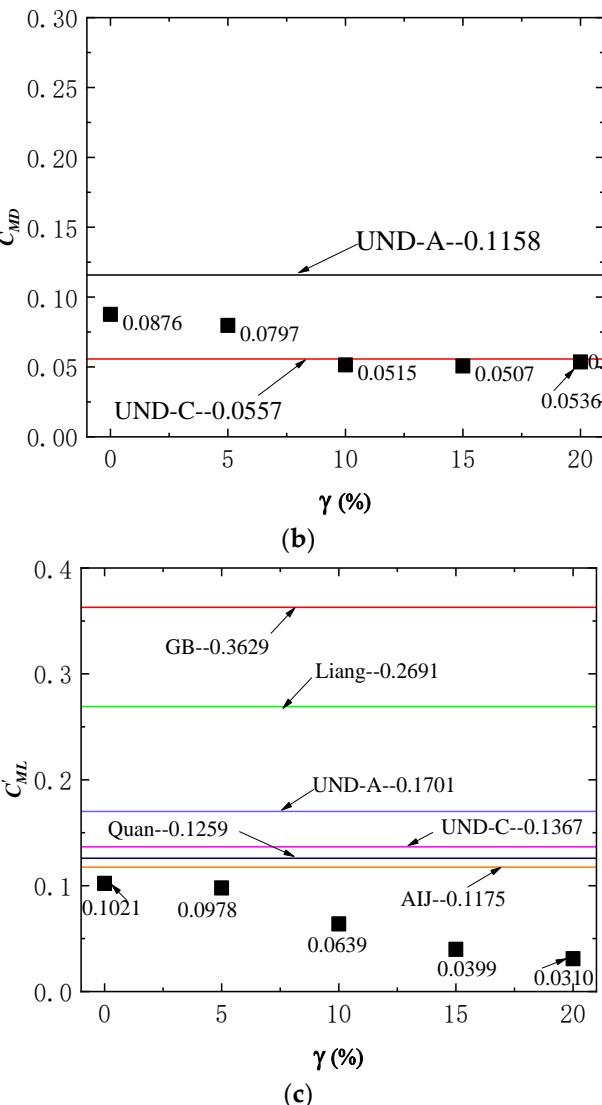

**Figure 11.** The base moment coefficients at different corner set-back rates.

3.3.2. Power Spectral Densities of Base Moment Coefficients

Figure 12 shows the power spectral densities of base moment coefficients. The trend of base moment power spectrum with reduced frequency is similar to the trend of the local wind force power spectrum with reduced frequency. The along-wind base moment power spectrum varies gentle with the reduced frequency, and they are insensitive to the corner set-backs. Significant changes in the across-wind base moment power spectra are observed at different corner set-back rates. Although all corner set-back models still show narrow-band characteristics in their power spectra, their peaks lower than those of the benchmark model, which is consistent with the variation of the power spectrum of lift forces. The 20% corner set-back model has a flatter across-wind power spectrum curve than the other three corner set-back models.

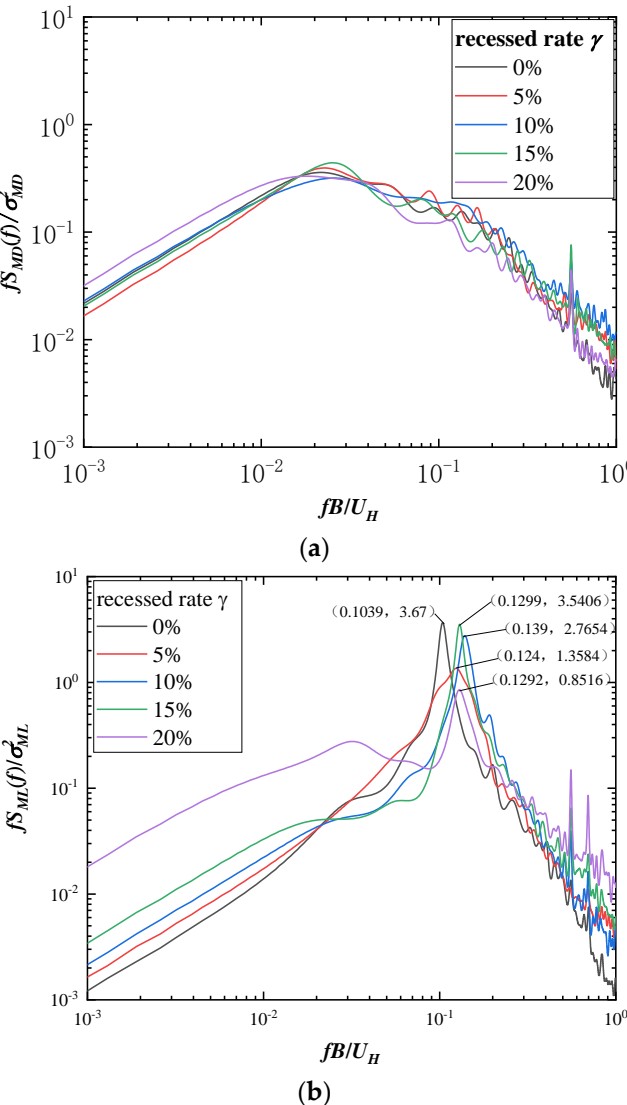

**Figure 12.** The base moment power spectra with reduced frequency. (**a**) The along-wind base moment power spectra. (**b**) The across-wind base moment power spectra.

## 4. Correlation Factors

Since the corner set-backs can affect the wind loads of rectangular high-rise buildings at varying extents, it is necessary to quantitatively assess the effects of different corner set-back rates. For this purpose, the correlation factors considering different corner set-back rates on the wind loads are proposed accordingly.

### 4.1. Correlation Factors for Base Moment Coefficients

The correlation factors for base moment coefficients are given by the following equations:

$$\eta_{MD}(\gamma) = \frac{C_{MD}(\gamma)}{C_{MD}(0)}, \ \eta\prime_{MD}(\gamma) = \frac{C\prime_{MD}(\gamma)}{C\prime_{MD}(0)} \tag{9}$$

$$\eta\prime_{ML}(\gamma) = \frac{C\prime_{ML}(\gamma)}{C\prime_{ML}(0)} \tag{10}$$

where $\eta_{MD}(\gamma)$ is the correlation factor of the mean base moment coefficients in along-wind. $C_{MD}(0)$ and $C_{MD}(\gamma)$ are the mean along-wind base moment coefficients without and with corner set-backs, respectively, $\eta\prime_{MD}(\gamma)$ is the correlation factor for the RMS base moment coefficients in along-wind. $C\prime_{MD}(0)$ and $C\prime_{MD}(\gamma)$ are the RMS along-wind base

moment coefficients without and with corner set-backs, respectively, and $\eta'_{ML}(\gamma)$ is the correlation factor of the RMS across-wind base moment coefficients. $C'_{ML}(0)$ and $C'_{ML}(\gamma)$ are the RMS across-wind base moment coefficients without and with corner set-backs, respectively. Since the mean across-wind base moment coefficient comes to nearly zero, it is not discussed in the paper.

Figure 13 displays the variations of correlation factors with different corner set-back rates for the mean along-wind base moment coefficients, RMS along-wind and across-wind base moment coefficients, respectively. With the increasing of corner set-back rate, the correlation factor of mean along-wind base moment coefficient decreases and then increases. The minimum correlation factor of the mean along-wind base moment coefficients is found when the corner set-back rate is 10%, which is about 0.52 times of the benchmark model. Regarding the RMS along-wind base moment coefficients, their correlation factors basically decrease by increasing the corner set-back rate. The minimum correlation factor is 0.58 times of the benchmark model. The correlation factors of the RMS base moment coefficients in across-wind also decrease by increasing the corner set-back rate. The minimum correlation factor reaches to 0.30 when the corner set-back rate is 20%.

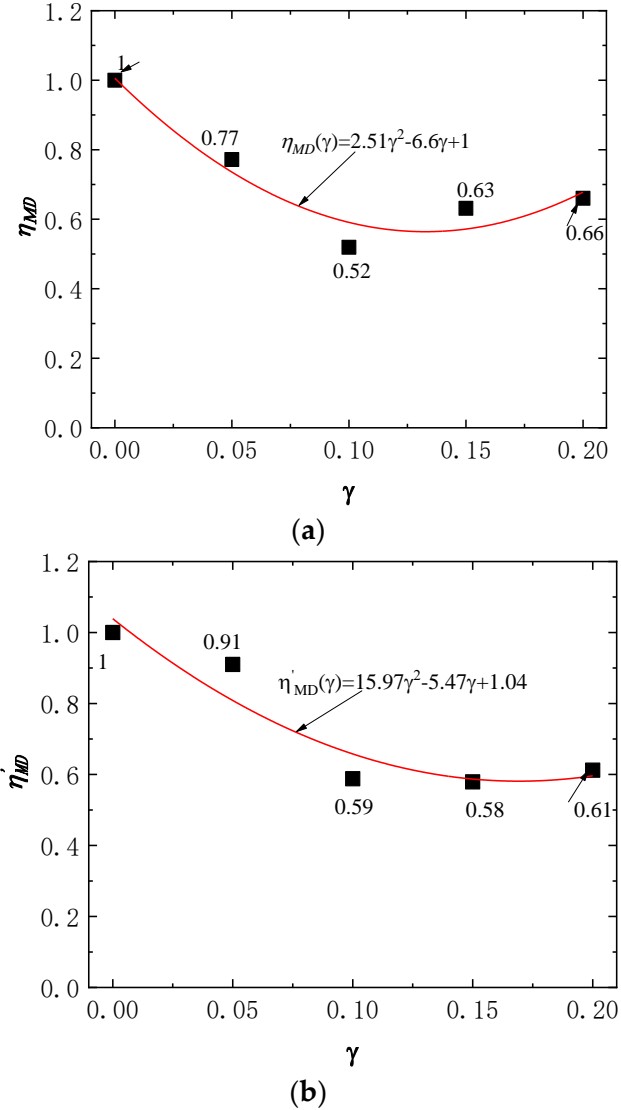

**Figure 13.** *Cont.*

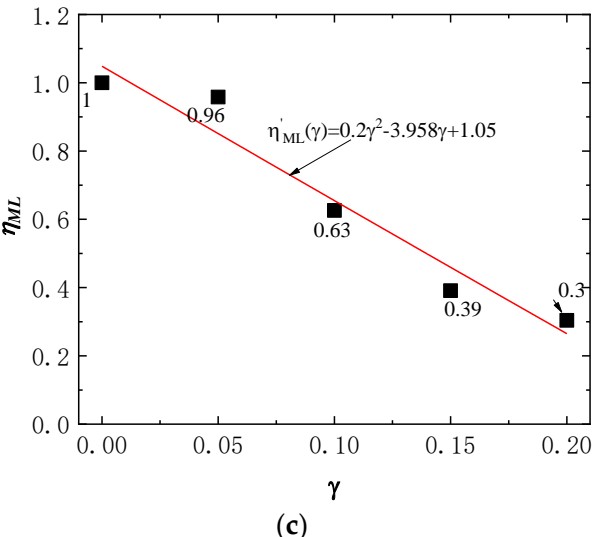

**(c)**

**Figure 13.** The correlation factors with different corner set-back rates. (**a**) The along-wind correlation factors for mean base moment coefficients. (**b**) The along-wind correlation factors for RMS base moment coefficients. (**c**) The across-wind correlation factors for RMS base moment coefficients.

By fitting the correlation coefficients of the base moment coefficients to obtain an empirical formula with the corner set-back rate as the independent variable, which offers a reference for engineering applications. The fitting formulas of correlation factors for the along-wind mean base moment coefficients, along-wind and across-wind RMS base moment coefficients are shown in Equation (11) to Equation (13). The fits of the test results to the corresponding formulas are 0.922, 0.895 and 0.949, respectively. In general, the fitting formula curves match well with the test results.

$$\eta_{MD}(\gamma) = 25.1\gamma^2 - 6.66\gamma + 1 \quad 0 \leq \gamma \leq 0.20 \tag{11}$$

$$\eta'_{MD}(\gamma) = 15.97\gamma^2 - 5.47\gamma + 1.04 \quad 0 \leq \gamma \leq 0.20 \tag{12}$$

$$\eta'_{ML}(\gamma) = 0.2\gamma^2 - 3.958\gamma + 1.05 \quad 0 \leq \gamma \leq 0.20 \tag{13}$$

*4.2. Correlation Factors for Power Spectral Densities of Base Moment Coefficients*

The correlation factors are defined as follow:

$$C_{m\,l}(f) = \frac{S_{M\,l}(f)}{S_{M\,0}(f)} \tag{14}$$

in which, $S_{M\,l}(f)$ and $S_{M\,0}(f)$ are the base moment spectrum with and without corner set-backs. The subscripts $l = D, L$ represent the along-wind and the across-wind, respectively. Figure 14 shows the correlation factors of the base moment spectra of models for a range of the reduced frequency from 0.10 to 0.25 [23]. It shows that the overall trend of correlation factor curves in corner set-backs is dispersed with reduced frequency. Compared with along-wind base moment power spectra, the correlation factors of across-wind base moment power spectra are fluctuated, especially in the range of 0.125 to 0.15. The maximum correlation factors for 10% corner set-back model and 15% corner set-back model are about 9.5 and 8.0, respectively. This phenomenon indicates that the increasing of the reduced frequency may lead to the magnification of the across-wind effects, which ought to attract the greater attention by structural designers of high-rise buildings. Table 1 lists the correlation factors in typical reduced frequencies ranged from 0.100 to 0.250, which serve as a direct reference for wind-resistant designs of similar tall buildings. In the Table 1, the along-wind correlation factors are mostly less than 1.0, and the optimal corner set-back rate is basically 20%. While the across-wind correlation factors are mostly greater than

1.0, this is means that when calculating the across-wind equivalent static loads of similar high-rise buildings considering corner set-backs by the stipulations in the Chinese [16] and Japanese [22] design code, the expanded correlation factors amplify the resonance coefficient and thus the dynamic wind loads in across-wind are amplified. As a result, wind engineers and structural designers ought to take this into account when assessing dynamic loads in across-wind for similar recessed high-rise buildings.

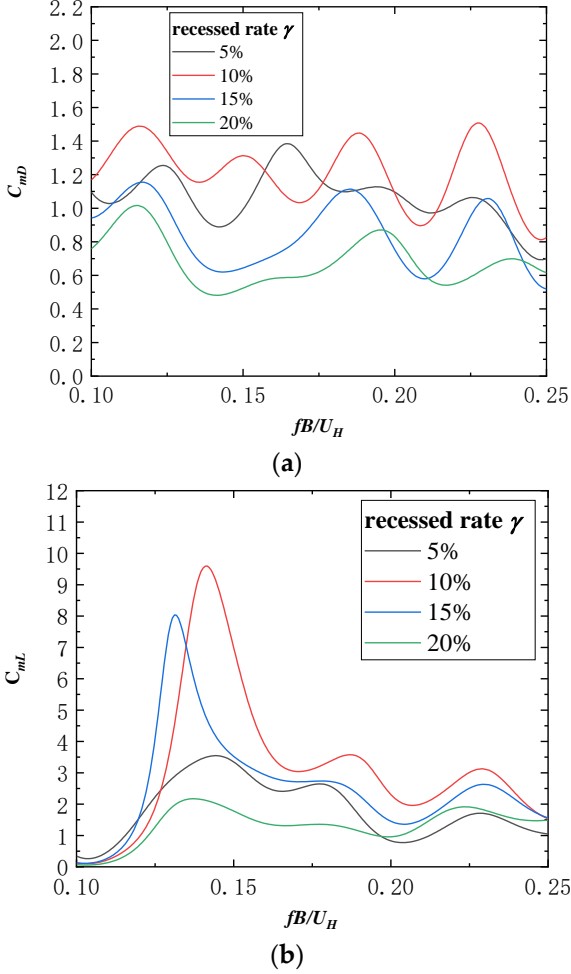

**Figure 14.** The correlation factors for base moment power spectra. (**a**) Along-wind. (**b**) Across-wind.

**Table 1.** The correlation factors for power spectral densities of base moment coefficients in typical frequencies.

| Wind Load Direction | Corner Set-Back Rate $b/B$ | Reduced Frequencies ($fB/U_H$) | | | | | | |
|---|---|---|---|---|---|---|---|---|
| | | **0.100** | **0.125** | **0.150** | **0.175** | **0.200** | **0.225** | **0.250** |
| along-wind | 5% | 1.095 | 1.249 | 0.994 | 1.179 | 1.097 | 1.063 | 0.701 |
| | 10% | 1.169 | 1.336 | 1.314 | 1.116 | 1.089 | 1.475 | 0.826 |
| | 15% | 0.941 | 1.005 | 0.644 | 0.955 | *0.753* | 0.953 | *0.519* |
| | 20% | *0.759* | *0.796* | *0.524* | *0.621* | 0.831 | *0.592* | 0.617 |
| across-wind | 5% | 0.340 | 2.230 | 3.345 | 2.617 | *0.827* | *1.656* | *1.049* |
| | 10% | 0.118 | 1.895 | 6.983 | 3.116 | 2.304 | 3.011 | 1.529 |
| | 15% | 0.126 | 4.105 | 3.522 | 2.728 | 1.452 | 2.508 | 1.559 |
| | 20% | *0.070* | *1.275* | *1.779* | *1.352* | 0.958 | 1.907 | 1.489 |

Notes: The numbers in italics and bold are the minimum values of the correlation factors of base moment power spectra.

### 5. Case Study

Prototype tall buildings with the same dimension of CAARC standard tall building (with a length of 45.72 m, a width of 30.48 m, and a height of 182.88 m) are selected to study the effects of corner set-backs on wind loads and wind-induced responses. The buildings are concrete structures, for which the densities are about 300 kg/m$^3$. The first two natural frequencies of the five models are assumed to be 2.743 s and 2.743 s, respectively [24]. When calculating wind loads, the design wind pressure is assumed to be 0.50 KN/m$^2$ with a damping ratio of 5% [16], while when calculating acceleration, the design wind pressure is assumed to be 0.35 KN/m$^2$ with a damping ratio of 2% [25–28]. Based on the linear mode [29], the first-order vibration mode function is $\phi(z_i) = (z_i/H)^{1.4}$, where $z_i$ is the height of the $i$ th floor from the ground, and $H$ is the height of the tall building. The tall buildings are evenly divided into 60 floors along the height and the height of each floor is H/60. Due to the greater impact (uncomfortable vibration) caused by wind-induced acceleration rather than wind-induced displacement for high-rise buildings, this study takes wind-induced acceleration of the top floor as the typical parameter to analyze the influences of corner set-backs on wind-induced response reduction of rectangular high-rise buildings. Due to the low efficiency and long calculation time of the time-domain method to calculate the structural wind load and response, and the results obtained by the time-domain method are not very convenient for engineering designers. Therefore, the frequency domain method is used to analyze the wind load and response of the structure in this paper.

In accordance with the theory of random vibration and structural mechanics and wind tunnel test results, the peak acceleration and RMS acceleration of the top floor can be determined by Equations (15) and (16).

$$\tilde{a}_l(z) = g\hat{a}_l(z) \tag{15}$$

$$\hat{a}_l(z) = \left( \int_0^\infty (2\pi n)^4 S_{Fl}(n)dn \right)^{\frac{1}{2}} \tag{16}$$

in which, the subscript $l$ can be taken as $D$ and $L$, representing the along-wind direction and the across-wind direction, respectively, $\tilde{a}(z)$ and $\hat{a}(z)$ are the peak acceleration and RMS acceleration at the height of $z$, respectively, $g$ is peak factor, this paper take 3.5, $S_{FD}(n)$ and $S_{FL}(n)$ is the along-wind and across-wind generalized force spectrum, respectively, and $n$ represents the natural frequency.

Figure 15 shows the values of the along-wind and across-wind peak acceleration at the top floor of the five different corner set-back high-rise buildings. Figure 15a shows that the peak acceleration in along-wind direction gradually decreases by increasing the corner set-back rate. The minimum peak acceleration 0.0477 m/s$^2$ is observed when the rate of corner set-back is 20%. In Figure 15b, the peak acceleration in across-wind direction decreases rapidly by increasing the corner set-back rate. The across-wind peak acceleration reaches the minimum value 0.0369 m/s$^2$ at the corner set-back rate of 20%.

The Chinese design load assumed that the RMS wind loads can be taken regard as the inertial wind loads and combine them with the mean wind loads to be the equivalent static wind loads for structural design of high-rise buildings. In this study, the equivalent static wind loads $\hat{F}(z)$, the mean wind loads $\overline{F}(z)$ and the inertial wind loads $\tilde{F}(z)$ are calculated using the below formulas:

$$\hat{F}_l(z) = \overline{F}_l(z) + \tilde{F}_l(z) \tag{17}$$

$$\overline{F}_l(z) = w_r \sum_{i=1}^N \overline{C_P}(i)A(i) \tag{18}$$

$$\tilde{F}_l(z) = gm(z_i)w_1^2\phi(z_i)\left( \int_0^\infty |H(n)|^2 S_{Fl}(n)dn \right)^{1/2} \tag{19}$$

where $w_r$ is the pressure at the height of $z$ (182.88 m), $m(z_i)$ is the mass of the floor at height of $z_i$, $w_1$ is the first modal frequency of the building, $\phi(z_i)$ is the coordinate of the mode shape, and $H(n)$ is the frequency response function.

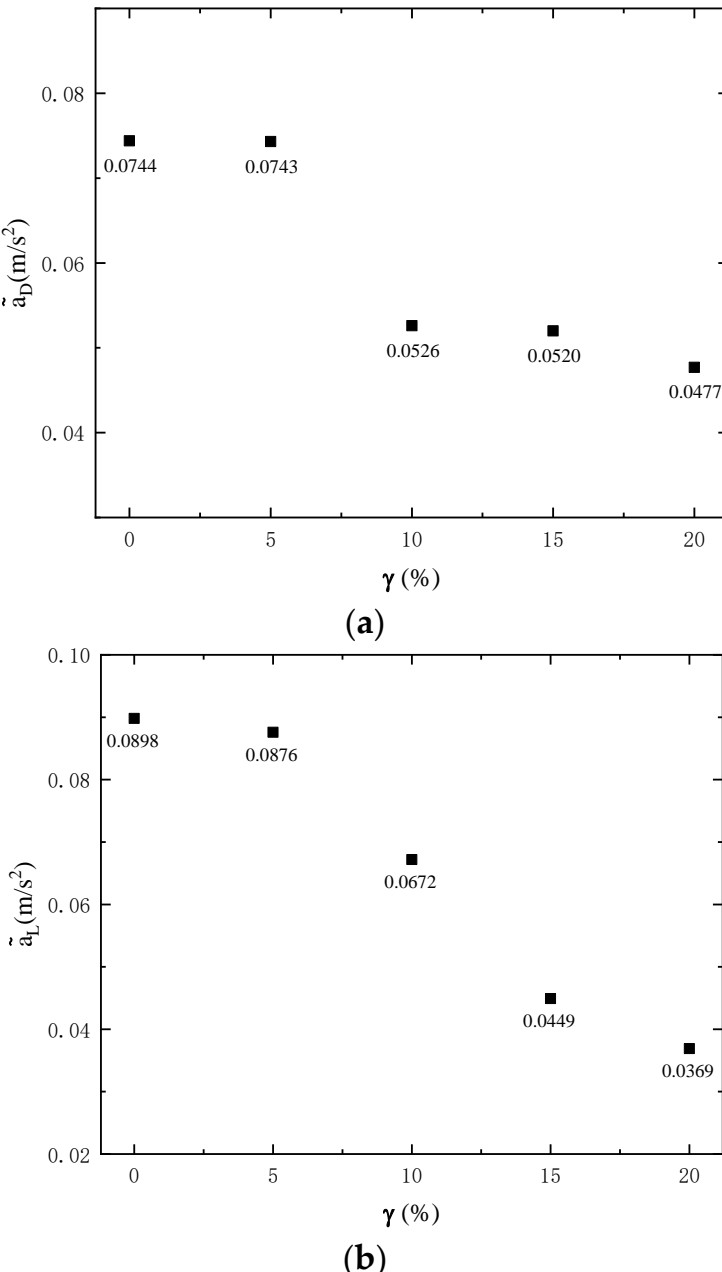

**Figure 15.** The peak acceleration at the top floor of the buildings. (**a**) Along-wind direction. (**b**) Across-wind direction.

Figure 16 illustrates the mean along-wind wind loads. The mean along-wind wind load of high-rise buildings decreases first and then increases with increasing of the corner set-back rates. The maximum reduction of mean along-wind wind loads happens at a corner set-back rate of 10%, and the overall along-wind mean wind loads is decreased by about 50%. This phenomenon is consistent with the variation trend of the mean drag force coefficients and mean along-wind base moment coefficients. As the mean across-wind wind loads are almost zero, and the across-wind mean wind loads is not discussed in this study.

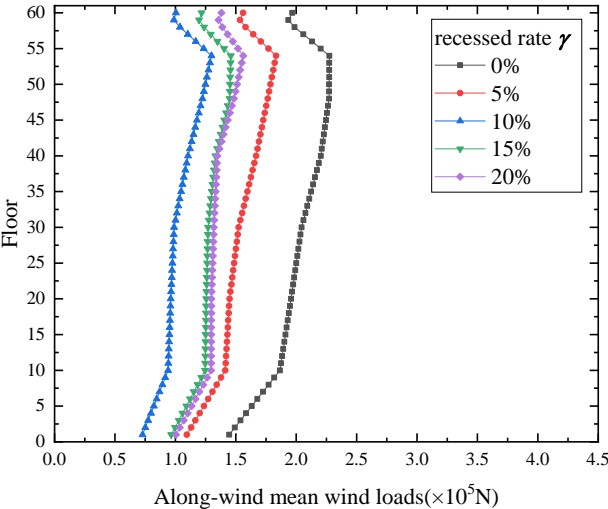

**Figure 16.** Along-wind mean wind loads.

Figure 17 represents the results of the inertial wind loads in along-wind and across-wind of high-rise buildings. In Figure 17a, the along-wind inertial wind load decreases by increasing the rate of corner set-back. The variation trend is consistent with the RMS drag force coefficients and RMS base moment coefficients in along-wind. When the rate of corner set-back exceeds 5%, the along-wind inertial wind load is basically constant with the increase of the corner set-back rate, and the along-wind inertial wind load on the top of high-rise buildings is reduced by approximately 40%. As seen in Figure 17b, inertial wind load in across-wind decreases with increasing of the corner set-back rate. The across-wind inertial wind load on the top of the building is decreased by about 67% for a corner set-back rate of 20%. In addition, the across-wind inertial wind load is greater than the along-wind inertial wind load when the corner set-back rate is less than 15%, while the along-wind inertial wind load is greater than the across-wind inertial wind load when the corner set-back rates are 15% and 20%.

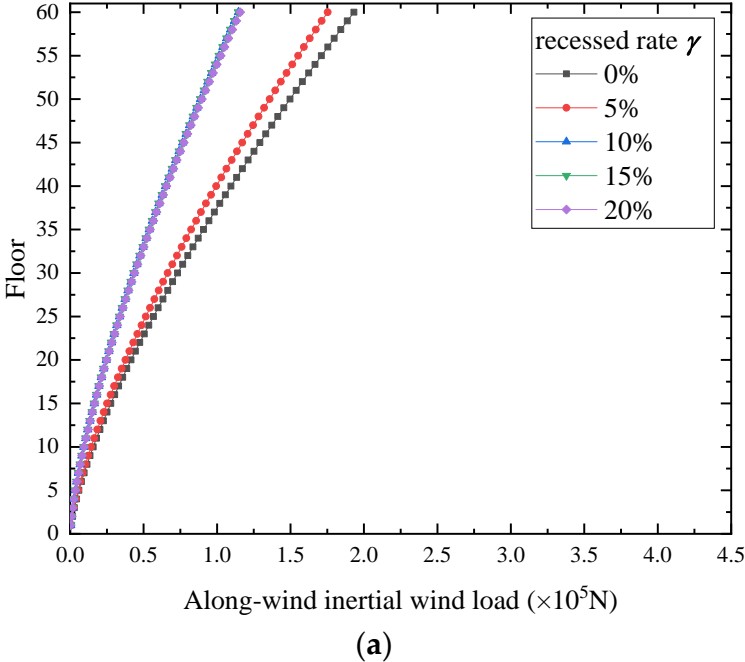

(**a**)

**Figure 17.** *Cont.*

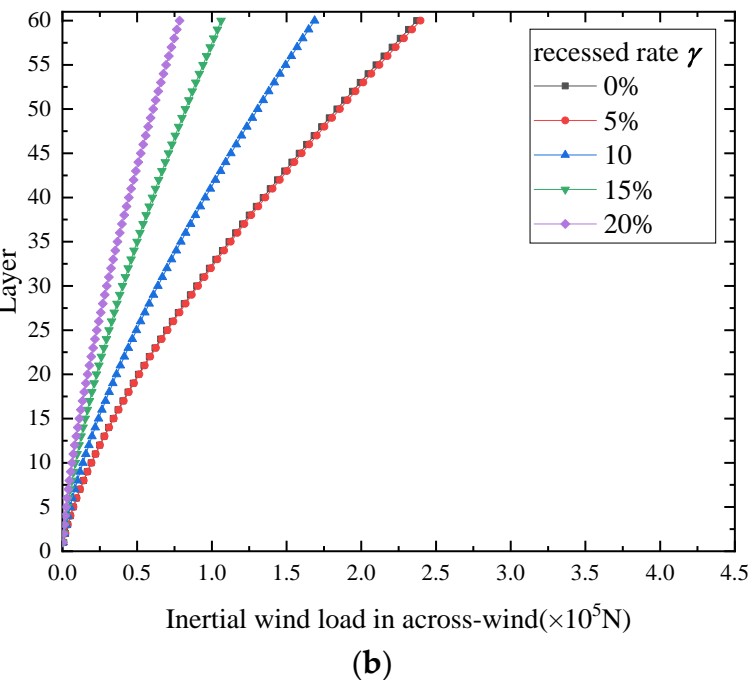

**(b)**

**Figure 17.** The inertial wind loads. (**a**) Along-wind inertial wind loads. (**b**) Across-wind inertial wind loads.

As the across-wind mean wind loads are close to zero at the unfavorable wind direction of 0°, the equivalent static wind loads in across-wind are equal to the inertial wind loads. Therefore, only the equivalent static winds at along-wind direction are presented and discussed in this study. Figure 18 displays the along-wind equivalent static wind load of high-rise buildings. The equivalent static wind load first decreases and then increases with the corner set-back rate increases. Once the rate of corner set-back exceeds 5%, the reductions caused by corner set-backs are almost the same for different rate of corner set-back including 10%, 15%, and 20%. The maximum reduction occurs at the corner set-back rate of 10%, and the overall loads is decreased by about 45%. In total, the corner set-back modification effectively reduces the equivalent static wind load in along-wind.

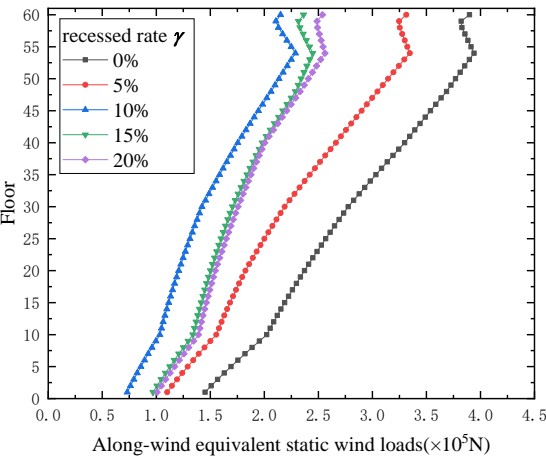

**Figure 18.** Along-wind equivalent static wind loads.

## 6. Conclusions

In this paper, wind effects on rectangular high-rise buildings with various corner set-back rates are investigated through a wide range of tests on five models. The distributions of wind pressure coefficients, local wind force coefficients, base moment coefficients and

the corresponding power spectrum are analyzed. Furthermore, correlation factors for wind loads reduction considering the effects of different corner set-back rates are discussed and the corresponding empirical formulas are proposed. The major findings of the paper are summarized below:

(1)  The absolute values of the mean wind pressure coefficients vary more significantly on the side walls with corner set-back rate. In addition, the RMS wind pressure coefficients on the side walls of the model with a corner set-back rate of 5% are greater than those of the benchmark model.

(2)  The local mean and RMS drag coefficient are reduced by corner set-back modification. The maximum reduction of the mean drag wind force is found when the rate of corner set-back is 10%. The RMS local lift coefficients for the 5% corner set-back model are greater than those of the benchmark model in the middle height.

(3)  The RMS across-wind base moment coefficients with various corner set-back rates are less than the values specified in the design codes.

(4)  The peak value for power spectral spectra of the lift force coefficients is reduced by corner set-back treatment and the 20% corner set-back model has a gentle curve of power spectra densities than other corner set-back models. However, the variation of the corner set-backs has little effect on the power spectra of drag force coefficients.

(5)  The correlation factors considering different rates of corner set-backs on the wind loads are proposed in accordance with the outcomes of wind tunnel test. The correlation factors in along-wind are mostly smaller within 1.0, while correlation factors in across-wind are greater than 1.0.

(6)  Based on the analysis of a case study, the peak of along-wind and across-wind acceleration is basically the same when the building with a corner set-back rate of 5% and without corner modification. The maximum reduction of along-wind mean wind loads is found when the corner set-back rate of 10%, and the overall along-wind mean wind loads is decrease by about 50%. The maximum reduction of across-wind inertial wind loads is found when the corner set-back rate of 20%, and the across-wind inertial wind loads of the top building is decrease by about 67%.

**Author Contributions:** Writing—original draft, Y.L.; Writing—review & editing, J.Y. and Y.Z. All authors have read and agreed to the published version of the manuscript.

**Funding:** Full support for the work was provided by the National Natural Science Foundation of China (Project Nos. 51708207, 51878271, 51978593 and 51778554) and the Natural Science Foundation of Hunan Province (Project No. 2020JJ5176).

**Institutional Review Board Statement:** Not applicable.

**Informed Consent Statement:** Not applicable.

**Data Availability Statement:** Not applicable.

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
