# Peer review of "Effects of Corner Set−Backs on Wind Loads and Wind Induced Responses of Rectangular Tall Buildings"

_applsci, doi:10.3390/app122412742_

Round 1
Reviewer 1 Report
The pressure measurements were carried out on a benchmark model (CARRC) and four models with different recessed rates (5%, 10%,15%,20%) in a boundary layer wind tunnel, and the results were analyzed. The research has a certain engineering application reference value and theoretical significance.
(1) The research and application of this manuscript should consider the influence of Reynolds number.
(2) In Figures 4 and 5, the units should be indicated
(3) In Figure 6, the height units should be marked
(4) Some type errors in the manuscript should be carefully checked. For example, the attach angle should be an attach angle.
(5) The manuscript should give the time domain information of the wind force and the frequency information.
Reviewer 2 Report
Review of the manuscript by Yi Li et al.
Effects of corner recessions on wind loads and wind induced responses of rectangular tall buildings
Submitted to Applied Sciences (2056829)
GENERAL COMMENTS
The paper deals with the effects of recessed modification on wind loads and wind-induced responses of rectangular high-rise buildings by wind tunnel tests. The study is presented in a fairly clear and concise way, with some useful findings. It seems that the outputs of this study aim to serve as references for wind-resistant design of similar buildings in strong wind region According to the high-quality standards of the Applied Sciences, the paper can be considered for publication after some revisions. Some major and minor comments are summarized as follows.
MAJOR/MINOR COMMENTS
u The INTRODUCTION is essentially a disordered list of concise (sometimes vague and imprecise) statements about what other authors did in the past in the field. The mere sum of these statements is far from a coherent analysis of the current state of the art and certainly does not suffice to support the paper’s motivations. The authors must relate the referenced works to each other by highlighting the advancements and significant theoretical/applied results of each piece of research. The motivation of the manuscript should be illustrated more clearly and concisely.
u English writing of the manuscript is not very good, please check it carefully and improve its quality.
u The sizes of figure a) and b) in Figure 1 are inconsistent with those of other figures in the whole text. Please adjust them to be consistent.
u Equation (1) is wrong, it should be , Please modify it.
u It is obvious that the volume of the building will be reduced after the corner recessions. Will the reduction of the volume affect the wind load and wind-induced response?
u I wonder you could change the horizontal coordinates of pictures 16, 17 and 18 to be the same?
u It seems that some conclusions are not new, and it is too wordy.
Reviewer 3 Report
1. Maybe "recessed" is not the ideal term. Corner set-backs maybe?
2. Figure 1-a and b, should be defined more complete...distribution of... along the height
3. Figure 2: why "simulated results". Are those values obtained from WT measurements?
4. The total number of pressure taps must be specified in the paper. Figure 7: why not representing the mean Cp's for all the pressure taps along the building height? The captions must be reformulated for completeness
5. It is not clear if the case study uses structural models to calculate de response using the WT data. This section must be significantly improved with clarifications and data. Why not automatically computing the wind loads from wind tunnel measurements and building response by running the time-history analysis?
6. Careful English and typing check of text is recommended
Round 2
Reviewer 3 Report
Linear-elastic time-domain analysis of building structures using wind tunnel measurements is effective since 30 years ago. The authors are kindly invited to investigate this major switch in the wind-induced response analysis of structures. In the last years, even the nonlinear response analysis became available as a major step forward in structural engineering.